# A Clinical Workflow for Evaluating Dose to Organs at Risk After Biology-Guided Radiation Therapy Delivery

**DOI:** 10.3390/cancers17243979

**Published:** 2025-12-13

**Authors:** Thomas I. Banks, Chenyang Shen, Andrew R. Godley, Yang Kyun Park, Rameshwar Prasad, Madhav Ravikiran, Shahed N. Badiyan, Tu Dan, Aurelie Garant, Robert Timmerman, Steve Jiang, Bin Cai

**Affiliations:** Department of Radiation Oncology, University of Texas Southwestern Medical Center, Dallas, TX 75390, USA

**Keywords:** biology-guided radiotherapy, PET linac, organ at risk dose evaluation

## Abstract

The RefleXion X1 is a novel linac designed to perform biology-guided radiation therapy (BgRT) by adjusting delivery of an external photon beam in response to real-time positron emission tomography (PET) signals from a ^18^F-fluorodeoxyglucose (FDG)-avid target. Due to the dynamic nature of BgRT, the delivered dose distribution may differ from the planned dose distribution. The RefleXion treatment planning system (TPS) can reconstruct and display the dose delivered in a BgRT treatment session upon its completion, but the TPS presents only a limited set of basic dose–volume metrics. We created a framework for calculating custom dose–volume metrics from the dose distribution provided by the RefleXion system, and for displaying the results in a trend plot which readily shows when the metrics are in violation of clinical dose tolerances. We incorporated this framework into a clinical workflow for systematically reviewing the doses received by organs at risk, and we retrospectively applied our approach to a selection of clinical cases. Our workflow can be useful for identifying changes of clinical concern in patient anatomy or target metabolic function, possibly before such changes adversely impact RefleXion BgRT delivery in subsequent fractions.

## 1. Introduction

The RefleXion X1 is a novel, closed-bore, binary multileaf collimator (MLC) linear accelerator (linac) designed to perform biology-guided radiotherapy (BgRT) [1,2,3] as well as standard image-guided radiotherapy (IGRT) [4]. The system’s BgRT capability, trademarked as SCINTIX technology (RefleXion Medical, Hayward, CA, USA), is currently FDA-cleared for treatment of ^18^F-fluorodeoxyglucose (FDG)-avid targets in lung or bone sites only [5], with some constraints on target size, range of motion, FDG activity, proximity to organs at risk (OARs), and treatment dose and fractionation. Expansion of SCINTIX to other treatment sites and/or using non-FDG radiotracers is an area of ongoing study [6,7,8].

SCINTIX therapy uses real-time positron emission tomography (PET) imaging to guide radiation dose delivery to ^18^F-FDG-avid targets even in the presence of target motion, with some limitations [9]. Due to the dynamic adjustment of the linac’s photon beam during SCINTIX treatment [2], the delivered dose distribution can differ from that in the original treatment plan. It is therefore desirable to have a means of estimating the doses actually delivered to organs at risk (OARs), as this would enable the clinical team to identify any toxicity violations as well as to monitor the stability of delivered dose from fraction to fraction.

The RefleXion treatment planning system (TPS) [10] provides a built-in feature for computing the dose delivered by a completed BgRT treatment fraction. This tool uses the imaging, linac, and couch data recorded by the system during a treatment session (i.e., machine log files) to calculate the dose deposited by the delivered beams on the patient’s planning CT, and it uses this reconstructed dose distribution to generate corresponding dose–volume histograms (DVHs) for all segmented structures. Although this dose reconstruction uses the beam data recorded during a dynamic SCINTIX treatment, it assumes the patient anatomy remains static for the entire course of treatment and therefore does not fully account for anatomical changes or patient motion during delivery. Nevertheless, the dose information calculated after delivery can be useful in detecting if unwanted intrafractional patient motion occurred [11] and in revealing interfractional changes in the targeted anatomy that are clinically relevant to nearby OARs and which could affect BgRT delivery.

In this paper we present the first report, to our knowledge, of an effort to integrate RefleXion’s post-BgRT dose calculation data into a routine clinical workflow. We focused on using the data to identify OAR dose deviations and trends of clinical concern during a course of treatment.

## 2. Materials and Methods

### 2.1. BgRT Platform

The RefleXion X1 system contains a 6 megavolt (MV) flattening-filter-free (FFF) linac, a binary multileaf collimator (MLC), an MV imager, a fan-beam kilovoltage CT (kVCT) imager for patient localization, and dual opposed scintillating PET imagers, all mounted on a rotating ring gantry [12]. The kVCT imager [13] is used to accurately position the patient before each IGRT or BgRT (SCINTIX) treatment, while the PET imager is used only for SCINTIX treatments. During a SCINTIX treatment the PET imager detects signals emanating from the region of the ^18^F-FDG-avid target and the system dynamically adjusts photon beam delivery in real time in response. The system also passes the target across treatment isocenter four times (via in-and-out passes of the couch) to guard against possible motion interplay effects and ensure the planning target volume (PTV) receives the full intended dose. There are several constraints on a target for SCINTIX treatment: the gross tumor volume (GTV) should be 1–5 cm in superior/inferior extent and always be at least 2 cm distant from FDG-avid OARs; PET activity in the PTV should be relatively uniform and must satisfy several metrics concerning signal strength; and the prescribed dose should be on the order of 8 Gy/fraction or larger, delivered in 5 or fewer fractions [2,5].

### 2.2. BgRT Delivery Workflow

The SCINTIX treatment workflow consists of the following steps [12,14,15]: conventional CT simulation of the patient; a functional-modeling (FM) session on the RefleXion machine to acquire PET data from the patient for use in treatment planning; treatment plan preparation in the RefleXion TPS (utilizing both the CT sim and FM images); kVCT-based patient localization at treatment, immediately followed by a PET “pre-scan” to verify that the target’s FDG avidity and extent have remained sufficiently stable; and finally, treatment delivery.

A unique aspect of SCINTIX plans created in the RefleXion TPS is the generation and use of bounded dose–volume histograms (bDVHs) for targets and OARs [15]. These are standard treatment plan DVHs but with bounds added to reflect the range of variations that could occur in actual treatment delivery due to changes in the target-to-background PET signal (±25%), patient setup shifts (5 mm in all directions), and dose uncertainty (±3%). The bDVHs are available for clinical evaluation of plan robustness prior to approval for treatment, and they are also used by the SCINTIX system itself when quantitatively evaluating the acceptability of the “pre-scan” PET distribution before allowing a treatment delivery to commence [2].

### 2.3. Built-In Tools for Post-BgRT Dose Calculation and Display

After a BgRT treatment session has been completed, the user can open the patient’s treatment plan in the RefleXion TPS (v2.1.29-4.3 in our case) and utilize the feature “Treatment Review > Compute Dose for Session” to calculate the dose delivered according to the system’s machine log files. With the RefleXion X1 this tool uses the planning CT image for its Collapsed Cone Convolution Superposition (CCCS) dose calculation [16] and thus does not fully account for changes to patient anatomy that may occur over the course of treatment. Also, the dose calculation is performed in the planning CT frame of reference (“patient point of view”) rather than the target frame of reference (“target point of view”) so in the case of a moving target the reconstructed dose distribution will be blurred to some degree. Since the patient anatomy is assumed to be static, the tool’s accuracy is limited when assessing the true dose received by any moving structure, whether a target or OAR.

Once the dose for a completed treatment session has been calculated, the TPS can display: the delivered and/or planned dose distributions for that fraction atop the planning CT images; the delivered-dose DVH curves for segmented structures atop the bounds from their respective planned bDVHs; and a default set of delivered-dose DVH metrics—Dmax, Dmin, Dmean, D98%, etc. (Figure 1). There are two notable restrictions on the post-BgRT dose viewer at present: it displays data for only a single fraction (as opposed to scaling up the results to the full course of treatment) and it cannot display any user-defined dose–volume metrics. Consequently, to perform a more rigorous quantitative evaluation of delivered dose it is necessary to do so outside of the RefleXion TPS.

### 2.4. Custom Dose Evaluation Framework

To better evaluate BgRT delivered-dose distributions, we implemented a framework that uses in-house software tools both to calculate custom OAR dose–volume metrics on the RefleXion-provided data and to graph the results by fraction.

Our first step is to export the RefleXion-provided session dose distribution as a DICOM dose file and then import this file into the Aria Eclipse TPS (Varian Medical Systems, Palo Alto, CA, USA; v16.01.00 in our case), where the dose is automatically associated with the same CT images used for planning in the RefleXion TPS (This association is straightforward because we actually prepare the CT images and segmented structures in Aria before sending them to the RefleXion TPS for planning, as the latter does not currently offer its own segmentation functionality). We then use the “Plan Sum” feature in Eclipse to create two different dose distributions: a scaled dose and a cumulative dose. A scaled dose is simply a session dose multiplied by the prescribed number of treatment fractions to scale it up to a full course of treatment, while a cumulative dose is created by adding up the dose from all delivered fractions plus the latest session dose for any remaining fractions. Cumulative dose distributions represent a reasonable prediction or record of total dose accumulation for the full course of treatment, but scaled dose distributions are more useful for monitoring delivery consistency because they amplify per-fraction variations that may be obscured in the cumulative dose. Note that simple cumulative-dose summations are possible because all post-delivery doses are calculated by the RefleXion TPS on the planning CT images and do not account for any changes in patient anatomy.

After a scaled dose distribution has been created, we run an established in-house Eclipse script to calculate custom dose–volume metrics from it and identify any OAR tolerance limit violations. This script has long been routinely used by our dosimetry team to document that every treatment plan satisfies physician-specified dosimetric objectives as well as standard OAR safety constraints before treatment, and its execution takes less than a minute. We enhanced the script expressly for this work to offer the option of exporting its calculated dose–volume metrics in Microsoft Excel format (Microsoft Corporation, WA, USA) for further processing. We created an Excel 365 template containing a Visual Basic script that graphs the OAR dose–volume metrics calculated for each delivered fraction (as well as for the original treatment plan), with each metric normalized to its respective tolerance limit for easy visual examination. In this way, as a patient’s treatment progresses we can readily identify OARs whose calculated metrics exceed tolerance limits or exhibit concerning trends. Our current framework is focused on OAR doses for two reasons: patient safety is paramount, and interpreting the dose received by the static contour representing a BgRT target can be a tricky proposition. It is conceivable a static structure could be defined that would serve as a surrogate indicator for the dose received by a moving BgRT target, but this is a topic requiring investigation.

### 2.5. Clinical Dose Evaluation Workflow and Its Retrospective Application to Patient Cases

We established a clinical workflow for systematically evaluating the dose delivered to OARs in a BgRT session, and we retrospectively applied this workflow to three of our previously treated patients as a test. Our workflow consists of two main parts: a qualitative review in the RefleXion TPS using its built-in features, followed by a quantitative evaluation using the custom software framework described above. A flowchart illustrating the workflow is shown in Figure 2.

After invoking the RefleXion TPS’s built-in “Compute Dose for Session” feature, we first examine the planned and delivered isodoses overlaid together atop the planning CT images to look for any obvious differences between them (Figure 1). For example, a noticeable offset between the two dose distributions might indicate the patient shifted during treatment delivery [11] or reflect a significant change in the targeted anatomy that occurred since the time of CT simulation. Next, we examine the delivered DVHs to ascertain if any plots deviate outside the planned bDVH bounds; any such deviation would suggest a change beyond what was deemed clinically acceptable, though this interpretation can be complicated in the case of a moving target. Any dose anomaly found in these examinations would provoke an investigation into its likely causes. This initial evaluation using only built-in features is largely qualitative because the RefleXion TPS does not extrapolate the session dose to a full treatment course, nor can it display user-defined dose–volume metrics.

After the qualitative review, we process and review the RefleXion dose data using our custom dose evaluation framework. This enables us to quickly identify if any of our user-defined OAR dose–volume metrics have worsened compared to the original treatment plan or previously delivered fractions.

We retrospectively applied our workflow to three previously treated BgRT patients to assess its clinical utility. Case 1: Ischium R (bone); the prescription was SBRT dose 3600 cGy in 3 fractions. The lesion was a secondary tumor ~3–4 cm in diameter. The full course of treatment was successfully delivered without any issues.Case 2: Lung LUL; the prescription was SBRT dose 5500 cGy in 5 fractions. The lesion was a primary tumor ~ 3 cm in diameter, abutting the anterolateral chest wall, and exhibited excursions of ~4 mm maximum in the superior/inferior direction during respiration. The full course of treatment was successfully delivered without any issues.Case 3: Lung LUL; the prescription was SBRT dose 6000 cGy in 5 fractions. The lesion was a primary tumor ~3.3 cm in diameter, located next to and partly infiltrating the anterolateral chest wall, with minimal motion (1–2 mm at most) during respiration. Treatment delivery was prevented at fraction 3 by SCINTIX interlocks.

Case 1 was selected as a representative static bone treatment. Case 2 was selected because it was a periodic-motion lung treatment similar to case 3 but had been delivered to completion. Case 3 was selected because it was the original motivation for the development of our workflow.

## 3. Results

Our retrospective qualitative reviews of the dose data for patient cases 1 and 2 in the RefleXion TPS were unremarkable. The results from our quantitative reviews of the two patients are presented in Figure 3. There we show the OAR dose–volume metrics evaluated on the dose distribution from each patient’s fraction 1 delivery (by way of example) scaled up to the full course of treatment, as well as trend plots of normalized OAR metrics from the original treatment plan and all delivered fractions. For case 1 all OAR metrics were stable and well below the tolerance limit boundary. For case 2 there were several OAR metrics that exhibited tolerance violations in the original treatment plan accepted by the physician, owing to their proximity to the lung target—namely, a rib passed through the PTV, and the relatively thin chest wall resulted in an elevated skin dose. The subsequent variations in these OAR metrics over the course of treatment could have been due to modest changes that were visible in the size and shape of the lung tumor, but we did not identify any clear correlation and ultimately there were no worsening trends of clinical concern.

In Figure 4 we present CT images and dose–volume results for patient case 3, whose full course could not be delivered via BgRT due to changes in the targeted anatomy. In this case the lung lesion was located at the lung/rib interface and shrank as treatment progressed. In our retrospective qualitative review of the planned and delivered isodose lines in the RefleXion TPS, some small lateral excursions (1–3 mm) into the chest wall were visible in fraction 1, but the deviations were not prominent. We did notice, however, that the DVH curves for the left brachial plexus and rib exceeded their respective bDVH boundaries near the maximum dose. For fraction 2, the delivered isodose lines showed significantly larger excursions into the chest wall (2–8 mm) and the DVH curves for the left brachial plexus and rib exceeded the bDVHs to an even greater degree. The increasing dose to these OARs is clearly evident in the normalized trend plot, and can be understood as a redirection of dose into the portion of the PTV containing rib. Brachial plexus toxicity can lead to severe injury, so in this case the increasing violations of the maximum dose limit on the left brachial plexus would have been concerning to the physician. The changes in the lesion’s PET activity accompanying its anatomical change ultimately triggered built-in safety controls of the SCINTIX delivery system which prevented delivery at fraction 3. A replan and reattempt to deliver fraction 3 also failed due to continuing tumor shrinkage, and the patient was ultimately moved to a conventional C-arm linac to complete the remainder of treatment.

## 4. Discussion

Our retrospective evaluation of dosimetric data from selected SCINTIX treatments supports our clinical experience that for most BgRT patients the dose is delivered as intended, owing to a combination of judicious treatment planning and the capabilities of the RefleXion platform. While the SCINTIX system performs a robust suite of internal system checks that has demonstrated effectiveness in catching problematic PET anomalies immediately before or during treatment, our post-delivery evaluation workflow could be helpful to users by highlighting OAR dose–volume changes and trends that may reveal developing issues of clinical concern or potential problems in future treatments.

The effectiveness of our workflow will vary from case to case, depending on a number of factors such as the relative locations of the target and OARs, the size of the OARs, target motion, the treatment prescription, the dose–volume metrics under consideration, etc. Patient case 1 was unremarkable and serves as a demonstration of a straightforward static-target scenario in which no OARs are in jeopardy. Our evaluation of patient case 2 illustrates that there can be relatively large variations in normalized OAR dose–volume metrics which are neither of clinical concern to the physician—despite being in violation of tolerance limits—nor appear to correlate with any notable changes in patient positioning or anatomy.

Patient case 3 was somewhat unique, and it was the motivation behind our development of the workflow described in this paper. The physicist supervising fraction 2 happened to notice the reduction in tumor size in the localization kVCT, prompting him to perform a careful follow-up. In reviewing the post-BgRT bDVHs he found the reported maximum doses to the left brachial plexus and rib had increased and exceeded our institutional tolerance limits. Although in this instance there was a visible change in tumor size and the supervising physicist caught it, we recognized that visual examination of kVCT images and isodose lines is not a reliable check. The RefleXion TPS did in this case show the left brachial plexus and rib maximum doses exceeding their respective boundaries in the recalculated bDVH plots, but we wanted a way to calculate and efficiently review the trends in our custom institutional metrics after every treatment fraction in order to better monitor the dose delivered to OARs. In this case the trend plot would have immediately revealed changes of concern, given the increasing trends of the left brachial plexus and rib dose–volume metrics as well as the violation of tolerance limits for some of those same metrics.

We have incorporated our dose evaluation workflow into routine clinical practice, performing it after every BgRT treatment session. If any concerning OAR dose deviations or trends are identified, we investigate, and notify the physician if warranted. By performing these checks while the course of treatment is still in progress we are better positioned to anticipate problems and respond accordingly. Our workflow could potentially warn of unnoticed changes in the target’s anatomy or PET activity which might lead to OAR toxicities or affect future treatment sessions.

We should emphasize that our primary goal in evaluating the post-BgRT dose distributions is to identify any changes of potential concern in the patient or the dose they are receiving, and from our perspective the segmented OARs offer the best means of doing so. When we apply our workflow in the clinic we are less concerned with the absolute accuracy of OAR delivered-dose metrics—which are reliable only for OARs which are not subject to motion and whose locations are unchanged from the planning CT—and are instead more focused on variations, especially those in or near the red tolerance-violation zone, that could indicate functional or anatomical changes have occurred in the target region and warrant closer scrutiny (e.g., by examining session kVCTs, PET images, etc.). The manner of follow-up will depend on the specifics of the situation but in some cases it might be deemed worthwhile to perform a new functional-modeling session and create a new treatment plan. We should also emphasize that our dose evaluation workflow does not provide any prescriptive guidance on interpretation of results or decision thresholds, which are left to the judgment of the user. The primary value we have added is the trend plot of custom OAR dose–volume metrics, as it serves as an informative dashboard and enables efficient review.

It is worth noting that the practice of using machine log files for post-delivery dose calculation is mature and well-established in radiation therapy at this point, and it is widely used for patient-specific IMRT quality assurance [17,18]. What is new here is its application in a BgRT setting, in which the dose delivered in each treatment session varies due to the real-time adjustment of beam delivery in response to the patient’s PET signal.

Given the limited set of features available in the RefleXion TPS for evaluating the dose delivered in a BgRT treatment session, we recommend that users supplement the system’s built-in capabilities with their own dose-processing framework. At present that is the only way to conduct customized dosimetric evaluation and graphing.

### 4.1. Limitations

The post-BgRT-delivery dose evaluation workflow we developed has limitations, some of which are inherent to the circumstances and some of which are self-imposed for practical reasons.

Most conspicuously, the workflow always utilizes the planning CT and static segmented structures for post-delivery dose computation and evaluation, and therefore it assumes the patient anatomy remains constant throughout the course of treatment. To perform more accurate dose reconstruction after a BgRT delivery session it would be necessary to instead utilize a kVCT localization scan, but this capability is not currently implemented in the RefleXion TPS.

Even if the patient anatomy were updated at each treatment session there would still be uncertainty in assessing the dose received by structures moving with respect to the static patient frame of reference. However, for most OARs the assumption of static anatomy should still yield a reasonable first-order estimate of the dose received and provide a means for monitoring constancy from treatment to treatment. For this reason we focused the quantitative portion of our workflow on identifying changes in the dose received by OARs.

### 4.2. Future Directions

We deliberately restricted the scope of this project to the development of a lean dosimetric evaluation workflow that could quickly provide clinically useful feedback on BgRT treatments. There are several ways in which we could expand or enhance the workflow to improve its accuracy, efficiency, and utility.

One potential enhancement would be to perform dose accumulation by utilizing the kVCT localization scans from treatment in conjunction with deformable image registration (DIR). The goal would be to perform each logfile-based dose reconstruction on a localization kVCT and then deform the kVCT to the planning CT. This would yield a more accurate cumulative dose record, but the kVCT-based dose reconstruction would need to be implemented by the vendor.

At present our workflow is somewhat labor intensive, requiring the user to transfer DICOM dose files between two different treatment planning systems, create scaled and cumulative dose distributions, run a script in the Eclipse TPS, and export and plot the resulting dose–volume metrics. We are seeking to automate this process as much as possible to reduce the user’s labor burden and enable them to focus their effort on interpreting results. Unfortunately the RefleXion TPS does not currently offer an application programming interface, so all data operations there (namely, initiation of dose reconstruction and its DICOM export) must be manually performed by the user via the graphical user interface.

As previously mentioned, the post-BgRT dose calculated for a moving target does not easily lend itself to meaningful interpretation. However, it is possible there is information to be gleaned about target motion and coverage from the doses received by surrounding segmented structures. For example, one could imagine a direct relationship between the degree of target motion and the doses received by nearby OARs, because more fluence is needed to deliver the full prescribed dose to a moving target. However, the reliability of this correlation would be case-specific and potentially complicated, being dependent on OAR proximity and the relative motion of target and OAR. We speculate that a more promising surrogate is the so-called biology tracking zone (BTZ), the fixed user-specified volume encompassing the target and required by SCINTIX therapy to limit the region from which PET signals are accepted and analyzed [2]. The percentage of BTZ volume receiving significant dose should correlate to the amount of target motion within it, so one would expect the magnitude and shape of the BTZ’s DVH curve to reflect the degree of target motion.

The workflow we presented here has been entirely focused on radiation dose and does not consider factors related to the PET radiotracer ^18^F-FDG, such as dosage parameters (e.g., injected activity and uptake time) or SCINTIX PET metrics acquired at the pre-scan or during treatment delivery. It is likely that changes of clinical significance in the patient will be reflected in changes to PET metrics or to the correlations between PET and dosimetric metrics. This is a potential area of investigation that involves complex interconnections between radiation dose and biology and merits deeper study. In the future we hope to incorporate PET metrics into our workflow.

## 5. Conclusions

We developed a simple workflow for evaluating the radiation dose calculated by the RefleXion TPS after a SCINTIX BgRT treatment session. Our workflow consists of a qualitative comparison of the planned and delivered fraction doses using features in the RefleXion TPS display, followed by a quantitative, graphical evaluation of custom OAR dose–volume metrics and trends using the Aria Eclipse TPS and in-house software tools. Our process could be helpful in clinical practice in revealing significant anatomical and functional changes which have occurred during the course of treatment and affected the doses received by OARs; the workflow also provides a record of cumulative delivered dose upon conclusion of treatment.

Future enhancements could include incorporating PET metrics, analyzing target dose, or using daily kVCT images in conjunction with deformable image registration to more accurately tally dose accumulation to OARs. We encourage users of SCINTIX technology to utilize the post-BgRT-session dose information from the RefleXion TPS to monitor for changes of concern in the patient that could otherwise go unrecognized.

## Figures and Tables

**Figure 1 cancers-17-03979-f001:**
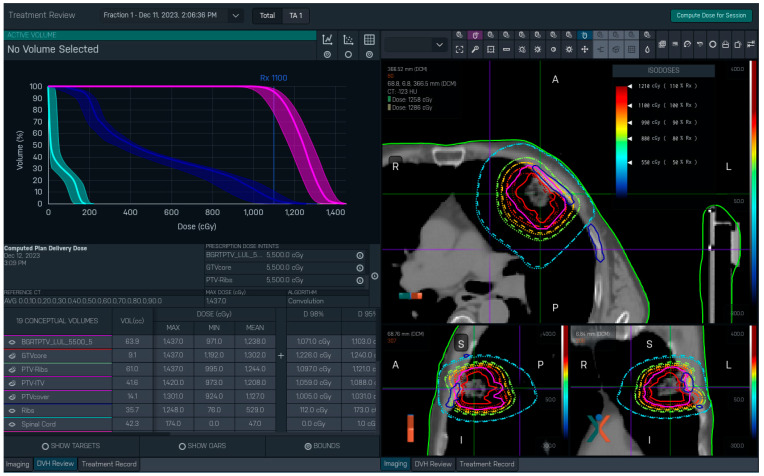
RefleXion TPS display of dose information calculated by the system after delivery of Fraction 1 for a BgRT lung treatment (patient case 2, discussed later). The planned and delivered dose distributions are overlaid on the planning CT scan on the right (planned = dotted lines, delivered = dashed lines). The solid lines correspond to the GTV (red), PTV (purple), and rib (blue) structures. Bounded DVH (bDVH) curves for selected structures (PTV, ribs, and spinal cord in this case) are displayed on the left. (Note: the DVHs show delivered curves atop planned bounds). A default set of dose–volume metrics are provided below the DVH curves.

**Figure 2 cancers-17-03979-f002:**
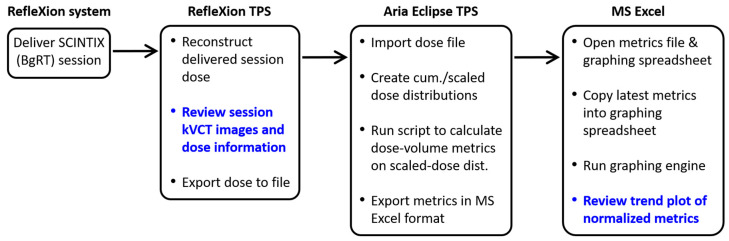
Flowchart of our clinical dose evaluation workflow. The data review steps are boldfaced.

**Figure 3 cancers-17-03979-f003:**
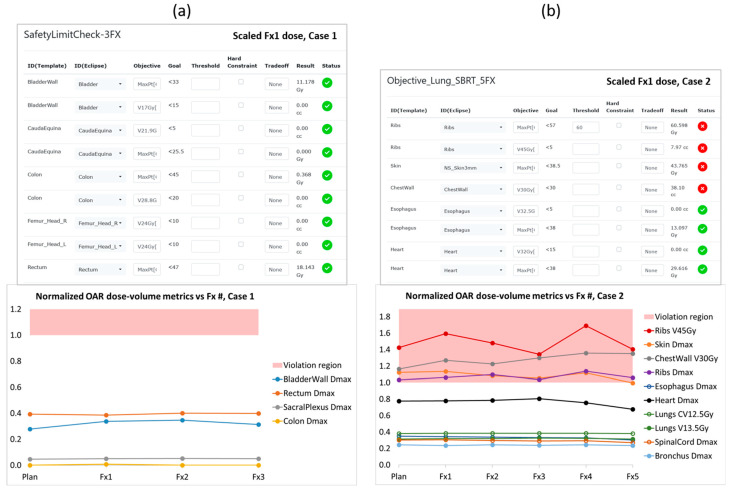
OAR dose–volume metrics lists and normalized trend plots from our retrospective evaluation of patient cases (**a**) 1 and (**b**) 2, which are described in the text. Each list of metrics was evaluated on the dose distribution calculated by the RefleXion TPS for the respective patient’s first BgRT fraction, scaled up to the full course of treatment. (Only partial lists are shown here for practical reasons). Dose-volume metrics that are within tolerance are indicated by a green circle with a checkmark, while metrics that exceed their respective tolerances are indicated by a red circle with an X. The OAR trend plots were created by calculating dose–volume metrics on the original treatment plan as well as the scaled dose distribution from every delivered fraction, and then normalizing each metric to its respective tolerance limit. Data points in the red shaded region are >1 and thus represent normalized OAR dose–volume results that have exceeded their tolerance limits.

**Figure 4 cancers-17-03979-f004:**
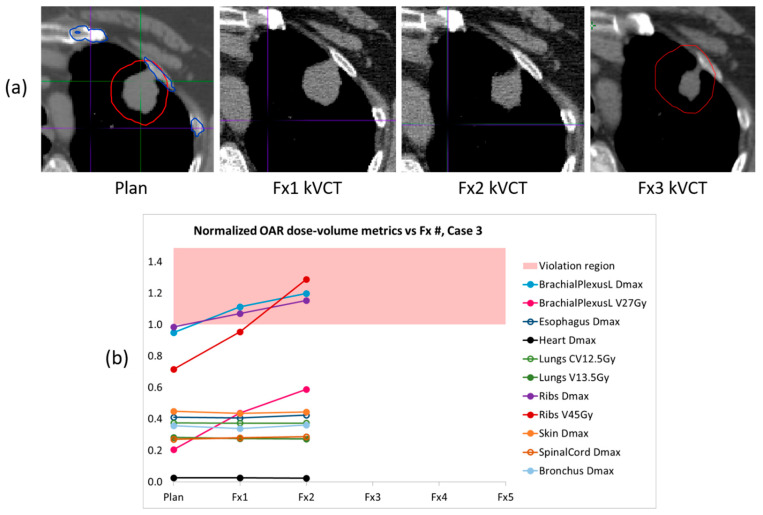
(**a**) Planning and localization axial CT images of the BgRT target lesion in patient case 3. The lesion was located at the lung/rib interface and the lung portion of the tumor shrank as treatment progressed. (**b**) The corresponding trend plot of normalized OAR dose–volume metrics shows increasing dose to the left brachial plexus and rib with each delivery, with some metrics in violation of their tolerance limits (red region). Fraction 3 could not be delivered via BgRT, so the patient received fractions 3–5 on a conventional linac.

## Data Availability

The original contributions presented in this study are included in the article. Further inquiries can be directed to the corresponding author.

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
