# Peer review of "A Clinical Workflow for Evaluating Dose to Organs at Risk After Biology-Guided Radiation Therapy Delivery"

_cancers, 2025, doi:10.3390/cancers17243979_

Round 1
Reviewer 1 Report
Comments and Suggestions for Authors
The manuscript entitled “Evaluation of dose to organs at risk after BgRT treatment delivery” presents a well-structured and highly relevant study. The authors detail the development and clinical implementation of a workflow for the systematic evaluation of organ-at-risk (OAR) doses following Biology-guided Radiotherapy (BgRT) treatments on the RefleXion® X1 PET-linac system.
The approach, which combines the RefleXion treatment planning system (TPS) for qualitative review with in-house software for quantitative analysis, provides valuable insights for clinical teams seeking to monitor OAR dose delivery accuracy. This is a timely and meaningful contribution to the operational understanding of post-treatment dose verification in the emerging BgRT paradigm.
I find the manuscript suitable for publication after moderate revision, as detailed below.
Major Comments
(a) The proposed workflow is sound and well-motivated, but its technical rigor would benefit significantly from a stronger quantitative component. The authors should provide evidence of the workflow's fundamental accuracy by comparing reconstructed dose values against an independent reference, such as a log-file–based verification or a dedicated phantom measurement.
(b) The retrospective application to only three clinical cases, while illustrative, restricts the generalizability of the findings. To consolidate the clinical relevance of the proposed workflow, the manuscript must include a concise summary table detailing key OAR dose metrics (e.g., Dmax, Dmean) across all three patients.
(c) Since the methodology heavily relies on institution-specific scripts and Excel tools, its reproducibility in other clinical settings is challenging. To facilitate adoption by other centers, the authors must provide a high-level flowchart or pseudocode that clearly illustrates the logical flow and the specific data exchange steps of the process.
(d) Although the authors focus on identifying OAR dose trends, the paper would be strengthened by briefly quantifying the typical range of observed variations. Please report the mean percentage deviation (or similar metric) between the planned and delivered OAR doses to provide context on the magnitude of the observed differences.
(e) While informative, the figures require improvement for enhanced readability. Please ensure the inclusion of numerical scales on all axes, utilize clearer and more consistent legends, and maintain a uniform visual style across all trend plots.
Minor Comments
- Correct all typographical errors (e.g., “worfklow” → “workflow”; adjust spacing errors, such as the one noted in the reference citation: “proceed[2]”).
- Specify all relevant software versions (TPS, in-house tools) and, if available, the typical execution times for the scripts used in the quantitative analysis.
- In the Discussion section, it would be highly valuable to comment on how this workflow could potentially interface with adaptive radiotherapy strategies or be automated using artificial intelligence (AI) tools for predictive dose monitoring.
- Clarify whether the analyses involving the “scaled dose” and “cumulative dose” have been previously clinically validated or are used exclusively for the retrospective evaluation presented in this study.
Author Response
Major Comments
(a) The proposed workflow is sound and well-motivated, but its technical rigor would benefit significantly from a stronger quantitative component. The authors should provide evidence of the workflow's fundamental accuracy by comparing reconstructed dose values against an independent reference, such as a log-file–based verification or a dedicated phantom measurement.
Validating the accuracy of the RefleXion system’s post-BgRT dose reconstruction is far beyond the scope of this study. The system’s machine log files are not available to the user, so phantom measurements would be necessary for such validation. We did perform extensive static and motion phantom measurements when commissioning our system for SCINTIX treatments, but our tests were focused on the accuracy of dose delivery to an FDG target, not on the larger dose distribution. We have seen no indications of systematic discrepancies between the planned and delivered BgRT dose distributions—they are generally consistent, and when we have found discrepancies there was an identifiable reason (eg, the tumor in patient 3 which shrank during treatment). Our focus in this manuscript is simply to report a practical method for evaluating the dose data provided by the RefleXion system, and provide some illustrative examples of its application.
(b) The retrospective application to only three clinical cases, while illustrative, restricts the generalizability of the findings. To consolidate the clinical relevance of the proposed workflow, the manuscript must include a concise summary table detailing key OAR dose metrics (e.g., Dmax, Dmean) across all three patients.
In this manuscript we are only reporting our motivation and method for evaluating delivered-dose data; we are not providing any prescriptive guidance on how interpret results or what actions to take. Interpretation will vary case by case depending on the specifics of the prescription (eg, total dose, number of fractions), the importance of nearby OARs, the relevant dose-volume metrics and their tolerances, etc. There are too many physician- and patient-specific variables involved to make any generalizations. We applied out workflow to three clinical cases as a way of testing it, and we present the results for the purpose of illustration.
(c) Since the methodology heavily relies on institution-specific scripts and Excel tools, its reproducibility in other clinical settings is challenging. To facilitate adoption by other centers, the authors must provide a high-level flowchart or pseudocode that clearly illustrates the logical flow and the specific data exchange steps of the process.
This request was made by multiple reviewers! We agree that a flowchart would be helpful to the reader, and we have added one (Figure 2) to better communicate the sequence of operations described in the text.
(d) Although the authors focus on identifying OAR dose trends, the paper would be strengthened by briefly quantifying the typical range of observed variations. Please report the mean percentage deviation (or similar metric) between the planned and delivered OAR doses to provide context on the magnitude of the observed differences.
We do not feel this would provide meaningful information for the reader. The numerical value of an OAR dose-volume metric can vary significantly from case to case, being highly dependent on a number of factors such as the relative location of the target and OAR, the size of the OAR, the treatment prescription, the dose-volume metric under consideration, etc. Indeed, this is why we chose to display the metrics on a normalized trend plot: it makes it possible to display a wide range of metrics on the same plot, and the user can see at a glance if any have violated their respective tolerance safety limits.
(e) While informative, the figures require improvement for enhanced readability. Please ensure the inclusion of numerical scales on all axes, utilize clearer and more consistent legends, and maintain a uniform visual style across all trend plots.
We thank the reviewer for bringing to our attention the lack of consistency in the visual style of the trend plots. We have improved the appearance of the trend plots and we are working to improve the readability of the figures by providing high-resolution versions with larger fonts.
Minor Comments
- Correct all typographical errors (e.g., “worfklow” → “workflow”; adjust spacing errors, such as the one noted in the reference citation: “proceed[2]”).
We thank the reviewer for identifying this typo. We have left the final formatting and spacing of citations to the journal.
- Specify all relevant software versions (TPS, in-house tools) and, if available, the typical execution times for the scripts used in the quantitative analysis.
We added version information on the RefleXion TPS, the Eclipse TPS, and Microsoft Excel, and we added text stating that the execution of the Eclipse script takes less than a minute.
- In the Discussion section, it would be highly valuable to comment on how this workflow could potentially interface with adaptive radiotherapy strategies or be automated using artificial intelligence (AI) tools for predictive dose monitoring.
Our institution is deeply involved in both adaptive radiotherapy and the application of AI to clinical radiotherapy, but our ambitions with this paper are far more modest! We are only seeking to report a practical clinical workflow which we think may be of interest to those involved in BgRT. We are reluctant to raise the topics of AI and adaptive RT for two reasons: (1) we are not privy to the internal target tracking data from SCINTIX, so we lack direct information about patient motion; and (2) we regard the OAR metrics calculated from the post-BgRT dose as rough estimates which nonetheless can be useful as an indirect way of catching changes in the patient of clinical concern. It is challenging to see how AI could be applied given our limited and imperfect view of the dose deposited in a BgRT delivery. Our workflow could be helpful in anticipating when it might be necessary to perform a new functional modeling session and create a new treatment plan, due to functional changes in the FDG-avid target; this would be an offline adaptive treatment flow, but it is not adaptive in the online sense as the term is generally applied. In short, we do not see any obvious avenues for adaptive RT or AI in this workflow and we are reluctant to idly speculate.
- Clarify whether the analyses involving the “scaled dose” and “cumulative dose” have been previously clinically validated or are used exclusively for the retrospective evaluation presented in this study.
To be clear, there is nothing novel in this portion of the workflow from a technical perspective. As we mention in the text, the creation of scaled and cumulative dose distributions in the Eclipse TPS is carried out using its built-in “Plan Sum” feature, which simply adds up dose distributions in a manner specified by the user. We also state in the text that the Eclipse script we use to calculate custom dose-volume metrics from the scaled/cumulative dose distributions is an established in-house tool used daily at our institution to evaluate clinical treatment plans. Both tools can be considered as having been thoroughly validated through initial TPS commissioning and longtime use.
If this comment is referring to the concept of using scaled and cumulative dose distributions for dose evaluation, that can be considered noncontroversial. A cumulative dose distribution is created by simply adding up the dose delivered at each fraction, while a scaled dose distribution represents the extrapolation of a single fraction to an entire course of treatment. As we mention in the text, this extrapolation magnifies any variations that occured in a particular delivery, and thus facilitates comparison of a single delivery to the original treatment plan as well as to other deliveries.
Reviewer 2 Report
Comments and Suggestions for Authors
Congratulations on this excellent work.
The use of PET has significantly improved oncology and radiation treatments: it is used as a predictive prognostic tool (https://doi.org/10.3390/cancers17183065) and as a tool to distinguish true progression from pseudoprogression (https://doi.org/10.3390/brainsci12040416).
This work demonstrates how it is possible to maintain the dose to the OARs and the target thanks to the new BgRT method.
It would be interesting, as a future development, to combine BgRT with SGRT (surface-guided RT) to further reduce the dose to the OARs and modulate the beam (on-off beam) on the PTV. Best regards
Author Response
We thank the reviewer for their complimentary feedback.
Reviewer 3 Report
Comments and Suggestions for Authors
Dear Authors, I send you my comments, please address it in your manuscript, which can enhance the readability, understanding and novelty of yours works.
Major Comments:
- Could the authors include a schematic illustration in the Introduction to visually explain the novelty and workflow of their study, especially showing how BgRT dose evaluation integrates with PET-guided real-time adjustments?
- How does the proposed workflow differ from or improve upon existing RefleXion® X1 TPS capabilities for post-treatment dose reconstruction and analysis?
- Have the authors validated their in-house quantitative analysis software or workflow against any established dose-evaluation tools or clinical benchmarks to confirm accuracy and reproducibility?
- Can the authors clarify how changes in patient anatomy or target metabolic function are detected and quantified within their workflow, and whether this has any implications for adaptive therapy planning?
- Would including a summary figure comparing planned vs. delivered dose distributions help readers better visualize the impact and novelty of the BgRT workflow?
Minor Comments:
- Please correct typographical errors such as “worfklow” and ensure consistency in capitalization and terminology (e.g., “BgRT,” “RefleXion® X1”).
- Can the authors provide more details about the patient selection criteria for the three retrospective BgRT cases to strengthen the clinical relevance?
- It would be helpful to clarify whether the spreadsheet-based OAR DVH analysis tool is publicly available or intended for internal clinical use only.
- Please expand briefly on future automation possibilities mentioned in the conclusion—what specific steps or parameters could be automated in the next iteration of the workflow?
- The Simple Summary could be made more concise and focused on clinical significance to enhance readability for a general scientific audience.
Best Regards
Comments on the Quality of English Language
Fine
Author Response
Major Comments:
Could the authors include a schematic illustration in the Introduction to visually explain the novelty and workflow of their study, especially showing how BgRT dose evaluation integrates with PET-guided real-time adjustments?
We added a flowchart of our workflow (Figure 2). This was requested by multiple reviewers and we agree this will help the reader to better follow the descriptions in the text.
To be clear, our dose-evaluation workflow can be applied only after a BgRT delivery is complete, because that is when the RefleXion TPS offers the option to compute the delivered dose. We acknowledge this point was not clear in the abstract, so we added some clarifying language there.
How does the proposed workflow differ from or improve upon existing RefleXion® X1 TPS capabilities for post-treatment dose reconstruction and analysis?
Our workflow utilizes the dose distribution reconstructed by the RefleXion system after a BgRT delivery, and we are not offering anything beyond that capability. Regarding its evaluation, we mention in the Methods section: “There are two notable restrictions on the post-BgRT dose viewer at present: it displays data for only a single fraction (as opposed to scaling up the results to the full course of treatment) and it cannot display any user-defined dose-volume metrics. Consequently, to perform a more rigorous quantitative evaluation of delivered dose it is necessary to do so outside of the RefleXion TPS.” In short, our workflow is designed to perform a more thorough evaluation of the post-delivery dose distribution by calculating and plotting custom dose-volume metrics.
Have the authors validated their in-house quantitative analysis software or workflow against any established dose-evaluation tools or clinical benchmarks to confirm accuracy and reproducibility?
Can the authors clarify how changes in patient anatomy or target metabolic function are detected and quantified within their workflow, and whether this has any implications for adaptive therapy planning?
To be clear, there is nothing novel in our workflow from a technical perspective. As we mention in the text, the Eclipse script we use to calculate custom dose-volume metrics from the scaled/cumulative dose distributions is an established in-house tool used daily at our institution to evaluate clinical treatment plans. It can be considered as having been thoroughly validated through initial TPS commissioning and longtime use.
Regarding the detection of changes in patient anatomy or target metabolic function: our workflow offers a possible way to detect such changes indirectly, through their effects on OAR dose-volume metrics. This approach will not always be successful in detecting such changes, as the results will be sensitive to the specifics of each clinical case, but it nevertheless offers users a way of monitoring the consistency of dose to OARs and will reveal if there is a violation of safety tolerances. When this happens, it is up to the user to investigate further into the cause—our method does not provide any guidance in that respect. In the “Future directions” subsection we mention our goal of incorporating the evaluation of PET metrics into our workflow at some point.
Would including a summary figure comparing planned vs. delivered dose distributions help readers better visualize the impact and novelty of the BgRT workflow?
We have attempted to provide an illustration of this sort in Figure 1, the screenshot of the RefleXion TPS post-delivery dose display. Generally speaking, the visual differences between the planned and delivered dose distributions are quite subtle, even in the case we report of the patient whose lung tumor shrank, increasing the dose delivered to nearby OARs. The fact that visual examination of the dose overlays is subjective was part of our motivation for creating a framework for quantitatively evaluating custom OAR dose-volume metrics.
Minor Comments:
Please correct typographical errors such as “worfklow” and ensure consistency in capitalization and terminology (e.g., “BgRT,” “RefleXion® X1”).
This has been done.
Can the authors provide more details about the patient selection criteria for the three retrospective BgRT cases to strengthen the clinical relevance?
Patient case 3 (LungLUL6000) was the original motivation for the development of this workflow, so it was natural to include that case. The other two cases were chosen from among our limited set of BgRT patients. We wanted a similar lung case that had been delivered to completion, which is how we selected patient case 2, and we wanted a bone case that had been delivered to completion, which is how we chose patient 1. We have added some information in this vein to the text.
It would be helpful to clarify whether the spreadsheet-based OAR DVH analysis tool is publicly available or intended for internal clinical use only.
We would be happy to share our spreadsheet with anyone who requests it. We will ask the editors how to best communicate that willingness in the article or its end matter, as we want to avoid cluttering the manuscript with statements about what is and is not shareable.
Please expand briefly on future automation possibilities mentioned in the conclusion—what specific steps or parameters could be automated in the next iteration of the workflow?
In the “Future directions” subsection of the Discussion we address this topic to some extent: “At present our workflow is somewhat labor intensive, requiring the user to transfer DICOM dose files between two different treatment planning systems, create scaled and cumulative dose distributions, run a script in the Eclipse TPS, and export and plot the resulting dose-volume metrics.” To answer the question above in more detail, the lowest-hanging fruit would be to write a master script to perform the entire sequence of operations which takes place in the Eclipse environment. The much larger challenge would be to write a program that interfaces with all three systems (RefleXion TPS, Eclipse TPS, and Excel) and can execute the entire workflow from beginning to end. This is not possible at present, however, and we have added this sentence to indicate why: “Unfortunately the RefleXion TPS does not currently offer an application programming interface, so all data operations there (namely, initiation of dose reconstruction and its DICOM export) must be manually performed by the user via the graphical user interface.
The Simple Summary could be made more concise and focused on clinical significance to enhance readability for a general scientific audience.
We agree the Simple Summary was lacking in focus and readability by a general audience, and we have made substantial revisions to improve it.
Reviewer 4 Report
Comments and Suggestions for Authors
Lines 126, 178, 187, 199, 200, 207, etc. throughout.: I notice throughout that the authors cite each case as what looks like the institutional nomenclature for ROIs in their TPS. I can readily guess what each one is, that LungLUL5500_5 is a left upper lobe lung tumor treated with 5500 cGy in 5 fractions; however, for a general readership, it is best to describe it in full. Of course, common acronyms like LUL are fine, it is just stylistically better to avoid using the institutional ROI coding (_underscores and all) and write out the description in full.
Lines 135-137: Please add some details to your three clinical cases somewhere in the “Materials and Methods” section. I realize that there’s some detail scattered throughout the paper, but there should be 1-2 paragraphs on Case description in the “Methods” section. Suggestions on what this content might look like:
- Where are the targets (bone can be anywhere, it matters where in the lung the targets are—mediastinal? lower lobes?);
- what are the prescriptions and fractionation?;
- what exactly do you mean by “uncomplicated”?; what do you mean by “presented difficulties during treatment”?
- do such difficulties bound what is “complicated” and what is “uncomplicated”?
- what is the primary disease?
I think that detail would give the reader better perspective as to what is being treated. E.g. there might be a case that is a lower left lung tumor, 3 cm in diameter, treated with an SBRT dose of 52 Gy in 4 fractions, of some pathology, and was complex as the target had a large motion trajectory of 3 cm due to proximity to the diaphragm. You can see how that level of detail can give the reader an appreciation of the clinical scenario, and even sell the reader on why the BgRT motion-compensation platform coupled with your novel workflow is clinically useful. Related to my previous comment on nomenclature: if you stated in the “Methods” what Case 1, Case 2, and Case 3 is, then you can refer throughout the paper to each Case#, rather than the coded institutional ROI nomenclature that is used throughout.
Lines 130-137, 2.4 Custom dose analysis workflow: In-house software based on the Varian environment is introduced here, meant to “process the dose information”. Considering that this is the key value-added feature that the authors are presenting (the title is “Evaluation of dose to organs at risk after BgRT…” after all), I believe more elaboration is required at this point. I realize that this elaboration of the in-house workflow/software occurs in the next paragraphs, which results in the next comment…
Lines 139-176: These paragraphs describe the qualitative evaluation using the built-in tools in the RefleXion TPS, and the quantitative evaluation using the Eclipse-based software. These paragraphs I believe belong not in the “Results” section, but rather in the “Materials and Methods” sections, specifically 2.3 and 2.4 (recall the immediately previous comment where it looked like detail on the in-house software was missing). While it is reasonable for some amount of “Methods” language to be included in the “Results” section (e.g. we weren’t expecting this quirk in this experiment while taking measurements, so we’ll describe it while showing the data), Lines 139-176 are very much describing the general method, and shouldn’t be in “Results”. This is a simple fix: just move these paragraphs into “Methods”, with some wordsmithing to smooth it over.
Figure 2: The text in the figure is far too small. A general rule of thumb is that the smallest text in a figure should be within at least ~3 points of font size of the main text; however, in Figure 2, the smallest text size is by my measurement 9 points of font size smaller than the main text. Even though this is a digital publication, the reader should be able to read at normal zoom levels. Moreover, Figure 2 is pixelated when zoomed in to the 500% required to read the smallest text, which further underlines that the text really should be reformatted. I realize that this is a screenshot of a TPS-related application, but there are ways of displaying this info without having such a small font size.
Figure 2 and 3: Both figures show LUL lung cases, yet Figure 2 shows 11 OAR metric trend lines while Figure 3 shows the “problem” lung case with only 3 OAR metric trend lines. As both are in the upper lung, you would expect the list of OAR metrics in play to be broadly similar, but they are not—why is this? If the complete OAR metric list was shown in Figure 3, would it obscure some of that upward trend, or add to it?
Lines 175-176: “that are ostensibly exceeding their dose-volume tolerances or trending in that direction.” What does “ostensibly” mean? Either tolerances are breached or they are not. What does “or trending in that direction” mean? How many fractions make a trend? How many OAR metrics make a trend that is a potential problem? How large does the variation have to be to make a trend that raises a flag?
Line 221: It is difficult to follow which lung case is being referred to here, as there are two lung cases. It would be far easier to follow along throughout if each of Case 1, Case 2, and Case 3 were described in a section in “Materials and Methods” (as I recommended previously), and then throughout the rest of the paper you can refer to each case by number.
Lines 221-233: The claim is “our method would have readily flagged” an issue with one of the lung cases. This paragraph then describes how the case was flagged: viewing the kVCT which indicated shrinkage, and follow-up review of the bDVHs. Both of these methods use features that are already in the RefleXion TPS application. Thus I do not see how the authors’ claim is valid, i.e. that their method would have caught this issue, if the features already in the TPS were adequate for screening. I take issue with the statements “for most patients it is more likely that changes in post-BgRT OAR doses could warn of unnoticed changes in the target’s anatomy or PET activity which might lead to toxicities or affect future treatment sessions”, this is a broad claim not supported by data in the article. The claim “our method would have raised warning flags as early as fraction 1, before there was a visible change in tumor size” is not factually supported in the text—how would it have raised a flag, and what part of your method would have flagged it? I might assume that it refers to the upward trend in Figure 3? If so, I do not see much difference in the % increase of the rib OAR metrics between Plan and Fx1 for the two lung cases in Fig 2 and 3—if this is the warning flag that the authors are citing, what’s the difference between the case where it delivered and the case where the course didn’t complete? The rib doses went up between Plan and Fx 1 for both lung cases, and had at least one rib metric in the red zone for Fx 1—wouldn’t this flag a problem for both lung cases?
Lines 239-243: This is the first mention of machine log files in the paper, but I suppose it was implied earlier on when calculating according to the post-BgRT “treatment record” (Line 107). I think mention of the term “machine log files” would be helpful at Line 107, as it would link up with the later mention in the Discussion.
Lines 244-253: “We are less concerned with the absolute accuracy of OAR delivered-dose values… and are instead more focused on variations which could indicate that functional or anatomical changes have occurred in the target region and warrant closer scrutiny”; and Lines 182-183: “make it easy to recognize if a particular OAR is violating its nominal tolerance limits and/or if its situation is worsening.” As with any QA/screening tool, there must be thresholds in order to enable the decision-making—the above quotes from the paper are quite ambiguous, it seems you are saying “both absolute dose accuracy and relative variation, one more than the other”. In the context of your work, if variations truly are the important thing (as you say in 244-253), you might set, say, a +20% increase relative to the reference plan, above which you would further investigate and below which you wouldn’t. Also, how many OARs need to raise a flag before you investigate? The claim is this is a “quantitative, graphical analysis of OAR dose metrics” (Lines 317-318)—such an analysis is incomplete without thresholds, and an algorithm to interpret the thresholds if there’s more than one. To recap, this is ambiguous: are the thresholds based on absolute OAR doses i.e. the “red zones”, or based on variation like is claimed in Lines 244-253, and if both, in what proportion, and navigated by what algorithm?
Lines 235-237: “If any concerning OAR dose deviations or trends…” Again, echoing my previous comment, this sounds like decision-making is based on “magnitude of concern”, which isn’t quantitative. Specific, threshold-based, decision-making criteria are needed to flag issues. Again, I understand that you have the “red zones” displayed in Figures 2 and 3; however, you also state that you are “more focused” (Line 249) on % increase of OAR metrics—these all need thresholds and a governing algorithm, since you are claiming this is a quantitative method.
Discussion: Only one of the lung cases was discussed, what of the other two cases? Shouldn’t there be interpretation applied to all the data in the Discussion? The Discussion section should link back to the data/observations in the “Results” section.
Author Response
Lines 126, 178, 187, 199, 200, 207, etc. throughout.: I notice throughout that the authors cite each case as what looks like the institutional nomenclature for ROIs in their TPS. I can readily guess what each one is, that LungLUL5500_5 is a left upper lobe lung tumor treated with 5500 cGy in 5 fractions; however, for a general readership, it is best to describe it in full. Of course, common acronyms like LUL are fine, it is just stylistically better to avoid using the institutional ROI coding (_underscores and all) and write out the description in full.
We thank the reviewer for pointing this out. We are so habituated to our institutional nomenclature that we failed to recognize it would be unfamiliar to the casual reader. We have removed this terminology and described the dose fractionation in full.
Lines 135-137: Please add some details to your three clinical cases somewhere in the “Materials and Methods” section. I realize that there’s some detail scattered throughout the paper, but there should be 1-2 paragraphs on Case description in the “Methods” section. Suggestions on what this content might look like:
Where are the targets (bone can be anywhere, it matters where in the lung the targets are—mediastinal? lower lobes?);
what are the prescriptions and fractionation?;
what exactly do you mean by “uncomplicated”?; what do you mean by “presented difficulties during treatment”?
do such difficulties bound what is “complicated” and what is “uncomplicated”?
what is the primary disease?
I think that detail would give the reader better perspective as to what is being treated. E.g. there might be a case that is a lower left lung tumor, 3 cm in diameter, treated with an SBRT dose of 52 Gy in 4 fractions, of some pathology, and was complex as the target had a large motion trajectory of 3 cm due to proximity to the diaphragm. You can see how that level of detail can give the reader an appreciation of the clinical scenario, and even sell the reader on why the BgRT motion-compensation platform coupled with your novel workflow is clinically useful. Related to my previous comment on nomenclature: if you stated in the “Methods” what Case 1, Case 2, and Case 3 is, then you can refer throughout the paper to each Case#, rather than the coded institutional ROI nomenclature that is used throughout.
We thank the reviewer for these helpful recommendations. We agree that labeling the retrospectively analyzed patient cases by number simplifies their presentation, and we have implemented that recommendation. We have also added more information about each case, per the suggestions above. We eliminated the terms “complicated/uncomplicated” and now simply state whether each treatment was delivered to completion or not.
Lines 130-137, 2.4 Custom dose analysis workflow: In-house software based on the Varian environment is introduced here, meant to “process the dose information”. Considering that this is the key value-added feature that the authors are presenting (the title is “Evaluation of dose to organs at risk after BgRT…” after all), I believe more elaboration is required at this point. I realize that this elaboration of the in-house workflow/software occurs in the next paragraphs, which results in the next comment…
Lines 139-176: These paragraphs describe the qualitative evaluation using the built-in tools in the RefleXion TPS, and the quantitative evaluation using the Eclipse-based software. These paragraphs I believe belong not in the “Results” section, but rather in the “Materials and Methods” sections, specifically 2.3 and 2.4 (recall the immediately previous comment where it looked like detail on the in-house software was missing). While it is reasonable for some amount of “Methods” language to be included in the “Results” section (e.g. we weren’t expecting this quirk in this experiment while taking measurements, so we’ll describe it while showing the data), Lines 139-176 are very much describing the general method, and shouldn’t be in “Results”. This is a simple fix: just move these paragraphs into “Methods”, with some wordsmithing to smooth it over.
We agree this is a more natural organization of the material, and we have implemented the reviewer’s suggestion. We moved a significant chunk of text from “Results” to “Methods,” and we modified/added subsections in “Methods” to better organize the information presented there. We believe these extensive changes have improved the explanation of the in-house components of our workflow, as requested in the comment re: lines 130-137.
Figure 2: The text in the figure is far too small. A general rule of thumb is that the smallest text in a figure should be within at least ~3 points of font size of the main text; however, in Figure 2, the smallest text size is by my measurement 9 points of font size smaller than the main text. Even though this is a digital publication, the reader should be able to read at normal zoom levels. Moreover, Figure 2 is pixelated when zoomed in to the 500% required to read the smallest text, which further underlines that the text really should be reformatted. I realize that this is a screenshot of a TPS-related application, but there are ways of displaying this info without having such a small font size.
We thank the reviewer for bringing this resolution issue to our attention. We have recreated the figure (which is now Figure 3) with better resolution. If this is unsatisfactory we will work with the editors to find a solution to improve its readability.
Figure 2 and 3: Both figures show LUL lung cases, yet Figure 2 shows 11 OAR metric trend lines while Figure 3 shows the “problem” lung case with only 3 OAR metric trend lines. As both are in the upper lung, you would expect the list of OAR metrics in play to be broadly similar, but they are not—why is this? If the complete OAR metric list was shown in Figure 3, would it obscure some of that upward trend, or add to it?
We thank the reviewer for pointing out this inconsistency in our presentation of the two LungLUL cases. For the problem case (patient case 3) we originally only graphed the three dose-volume metrics which cross into the red violation region, because that is the behavior that would have triggered an investigation. However, the reviewer is absolutely correct that for the sake of consistency we should include the same metrics that are graphed for LungLUL patient case 2. We have recreated the trend plots for patient cases 2 and 3, and we have applied a consistent labeling scheme for the metrics in both.
Lines 175-176: “that are ostensibly exceeding their dose-volume tolerances or trending in that direction.” What does “ostensibly” mean? Either tolerances are breached or they are not. What does “or trending in that direction” mean? How many fractions make a trend? How many OAR metrics make a trend that is a potential problem? How large does the variation have to be to make a trend that raises a flag?
Re: “ostensibly”: We used this qualifier to reflect the fact we have no way of knowing the accuracy of the calculated OAR dose-volume metrics, because BgRT is a dynamic process that responds to target motion but the post-BgRT dose computation is performed on a static CT image. It is possible, therefore, that a calculated OAR dose-volume metric may show a tolerance violation even if in reality the OAR did not actually receive such a dose—ie, it could be a false positive. (The converse could happen as well, a false negative.) However, we acknowledge that using the term “ostensibly” may only create confusion, so we have reworded the relevant passages in the abstract and text to simply state, eg, “OARs whose calculated metrics exceed tolerance limits.”
The judgment of what constitutes a concerning OAR dose violation or trend is left entirely to the user. We have added text to explicitly state that we are not providing prescriptive guidance on interpreting the numerical results from our workflow. (For me, any new tolerance violation, or a two-fraction trend approaching/crossing the tolerance limit, would be an investigation trigger, but that is solely my own opinion.)
Line 221: It is difficult to follow which lung case is being referred to here, as there are two lung cases. It would be far easier to follow along throughout if each of Case 1, Case 2, and Case 3 were described in a section in “Materials and Methods” (as I recommended previously), and then throughout the rest of the paper you can refer to each case by number.
We agree this would simplify the presentation of the retrospectively evaluated cases, and we have implemented the reviewer’s recommendation.
Lines 221-233: The claim is “our method would have readily flagged” an issue with one of the lung cases. This paragraph then describes how the case was flagged: viewing the kVCT which indicated shrinkage, and follow-up review of the bDVHs. Both of these methods use features that are already in the RefleXion TPS application. Thus I do not see how the authors’ claim is valid, i.e. that their method would have caught this issue, if the features already in the TPS were adequate for screening. I take issue with the statements “for most patients it is more likely that changes in post-BgRT OAR doses could warn of unnoticed changes in the target’s anatomy or PET activity which might lead to toxicities or affect future treatment sessions”, this is a broad claim not supported by data in the article. The claim “our method would have raised warning flags as early as fraction 1, before there was a visible change in tumor size” is not factually supported in the text—how would it have raised a flag, and what part of your method would have flagged it? I might assume that it refers to the upward trend in Figure 3? If so, I do not see much difference in the % increase of the rib OAR metrics between Plan and Fx1 for the two lung cases in Fig 2 and 3—if this is the warning flag that the authors are citing, what’s the difference between the case where it delivered and the case where the course didn’t complete? The rib doses went up between Plan and Fx 1 for both lung cases, and had at least one rib metric in the red zone for Fx 1—wouldn’t this flag a problem for both lung cases?
We thank the reviewer for pointing out some sloppiness in our arguments here. The reviewer is correct that we were making some broad claims (eg, “for most patients[…],” “our method would have raised warning flags as early as fraction 1”) that are not supported by the evidence presented in the manuscript. We have revised our wording in those cases to be more accurate.
Re: the lung patient with shrinking tumor (case 3): the reviewer is correct that the shrinkage was noticed and evaluated using existing features of the RefleXion TPS. However, this was thanks to the astuteness and initiative of the supervising physicist, who noticed the physical change and decided to perform a manual investigation and calculations which revealed a violation of the brachial plexus Dmax had occurred. With our new workflow, these calculations can be performed and displayed for the entire suite of custom OAR metrics of interest, and there is far less risk of missing a violation or trend because of observer variation. We recognize that the RefleXion display does have value, though, which is why our workflow consists of both a qualitative review in the RefleXion TPS and a quantitative review with our in-house software.
Regarding the OAR trend plots for the two lung cases: the reviewer is correct that the two-fraction trend of increasing doses in case 3, plus the entry of three separate metrics into the red violation zone, would have been a clear indication to us that further investigation was warranted. In contrast, with case 2 no metrics entered the red violation zone that were not already there in the original treatment plan. However, the reviewer is also correct that there is potentially some element of coincidence to the fact that the case which encountered delivery issues is also the one in which OAR metrics transitioned into the red violation region. One can imagine a scenario in which a metric started in the violation region and kept trending worse and worse. The situation will vary case to case, which is why we leave interpretation to the user, and why it is hard to present a comprehensive algorithmic solution. Our goal here is to take potentially useful dose information which might otherwise go unused and present it in an efficient manner.
Lines 239-243: This is the first mention of machine log files in the paper, but I suppose it was implied earlier on when calculating according to the post-BgRT “treatment record” (Line 107). I think mention of the term “machine log files” would be helpful at Line 107, as it would link up with the later mention in the Discussion.
We thank the reviewer for this suggestion, which has been implemented.
Lines 244-253: “We are less concerned with the absolute accuracy of OAR delivered-dose values… and are instead more focused on variations which could indicate that functional or anatomical changes have occurred in the target region and warrant closer scrutiny”; and Lines 182-183: “make it easy to recognize if a particular OAR is violating its nominal tolerance limits and/or if its situation is worsening.” As with any QA/screening tool, there must be thresholds in order to enable the decision-making—the above quotes from the paper are quite ambiguous, it seems you are saying “both absolute dose accuracy and relative variation, one more than the other”. In the context of your work, if variations truly are the important thing (as you say in 244-253), you might set, say, a +20% increase relative to the reference plan, above which you would further investigate and below which you wouldn’t. Also, how many OARs need to raise a flag before you investigate? The claim is this is a “quantitative, graphical analysis of OAR dose metrics” (Lines 317-318)—such an analysis is incomplete without thresholds, and an algorithm to interpret the thresholds if there’s more than one. To recap, this is ambiguous: are the thresholds based on absolute OAR doses i.e. the “red zones”, or based on variation like is claimed in Lines 244-253, and if both, in what proportion, and navigated by what algorithm?
Lines 235-237: “If any concerning OAR dose deviations or trends…” Again, echoing my previous comment, this sounds like decision-making is based on “magnitude of concern”, which isn’t quantitative. Specific, threshold-based, decision-making criteria are needed to flag issues. Again, I understand that you have the “red zones” displayed in Figures 2 and 3; however, you also state that you are “more focused” (Line 249) on % increase of OAR metrics—these all need thresholds and a governing algorithm, since you are claiming this is a quantitative method.
We thank the reviewer for pointing out that our terminology here was confusing. As discussed in some of our responses above, this manuscript is only reporting the procedure we developed for calculating and displaying custom dose-volume metrics after BgRT delivery, and how review of that information could be clinically useful when applied to OARs. We therefore agree that our use of the word “flag” was misleading, as it suggests application of an algorithm with decision thresholds, which is not the case. We have removed all instances of “flag”/”flagging” and reworded the surrounding text.
We feel it is still accurate, however, to refer to our workflow as “quantitative,” because we are calculating dose-volume metrics from the RefleXion-provided post-delivery dose distribution. We also feel it is appropriate to provide some modest commentary on what we ourselves consider important when we review the dose data clinically, in light of its limitations/shortcomings. But ultimately all judgments are up to the user.
We have revamped much of the Discussion text to clarify these many points. We thank the reviewer again for their feedback, which has resulted in vast improvements to the manuscript.
Discussion: Only one of the lung cases was discussed, what of the other two cases? Shouldn’t there be interpretation applied to all the data in the Discussion? The Discussion section should link back to the data/observations in the “Results” section.
We considered the workflow results for the other two cases largely unremarkable, which is why we stated that our retrospective investigation “supports our clinical experience that for most BgRT patients the dose is delivered as intended.” We felt it would be more illuminating to discuss the motivation behind our workflow (i.e., case 3) as well as the goals and potential benefits of a custom dose-evaluation framework.
Round 2
Reviewer 1 Report
Comments and Suggestions for Authors
Dear Authors, thank you for submitting the revised version of your manuscript and for carefully addressing the comments from the previous review. The changes you implemented have substantially improved the clarity and completeness of the work. The manuscript is now more coherent overall, the rationale for the proposed workflow is better articulated, and the presentation of the clinical cases is clearer and more effective.
After reviewing the revision, I believe that the manuscript is very close to being ready for publication. Only a few minor points remain that could further enhance clarity and presentation. These do not affect the scientific validity of the work but would help strengthen the final version.
First, the explanation of the scaled-dose evaluation could be expanded slightly. Although the concept is already described, adding a brief clarification on why the scaled dose is particularly useful for detecting fraction-to-fraction variations would help readers who may be less familiar with this approach. For example, in Section 2.4 you might consider modifying the sentence “A cumulative dose is not as sensitive to delivery changes as a scaled dose distribution because the latter magnifies variations that occurred in a single treatment session” to something along the lines of “Scaled dose distributions were especially useful for monitoring delivery consistency, as they amplify even small session-to-session variations that may otherwise be obscured when examining cumulative dose.”
Second, the discussion regarding the difficulty of interpreting target dose could benefit from a short connecting remark about potential surrogate indicators, such as the biology tracking zone (BTZ). You already elaborate on this concept later in the manuscript; introducing it briefly earlier in Section 4.2 would improve continuity. For instance, after noting the limitations of interpreting dose to a moving target, you could add: “Although direct interpretation of target DVHs is challenging under these conditions, surrogate indicators such as the dose distribution within the biology tracking zone (BTZ) may offer complementary insight and merit further investigation.”
The readability of the trend plots in Figures 3 and 4 could also be enhanced. The information they present is valuable, but the current contrast and density make them somewhat difficult to interpret quickly. Clarifying in the legend that the red region corresponds to tolerance violations, or enlarging the labels for the plotted metrics, would improve usability.
A few sentences could be tightened for clarity. In the Abstract, the sentence “Our workflow makes efficient use of existing RefleXion and in-house software resources” might be clearer if revised to: “The workflow integrates existing RefleXion tools with our in-house software to provide an efficient framework for post-delivery dose evaluation.” Similarly, in Section 3, the informal expression “nothing attention-grabbing” could be replaced with the more formal “no prominent deviations were observed.”
Lastly, I suggest considering the following revised title, which more clearly reflects the focus and contribution of the work: “A Clinical Workflow for Post-Delivery OAR Dose Evaluation in Biology-Guided Radiotherapy (BgRT)”
These refinements are minor and primarily editorial in nature. The scientific content is strong, and the manuscript represents a meaningful and practical contribution to the clinical implementation of BgRT.
I look forward to seeing the final version.
Author Response
[2nd author response] First we want to present some introductory remarks that we meant to include in our first response but which were inadvertantly omitted. These comments are still germane to our latest responses:
We wish to thank the reviewers for their careful reading of our manuscript and for their insightful comments. Their feedback made us realize we needed to firm up our report in many places and use clearer and more precise language throughout. We consider the revised manuscript to be a vast improvement on the original submission.
A number of comments from multiple reviewers indicated to us that our terminology may have created some underlying confusion about the nature and goals of our work. In particular, our use of the term “analysis” to describe our workflow may have been misleading. To be clear, we are not providing any prescriptive guidance on how to interpret post-delivery dose-volume metrics, nor are we providing any action thresholds—that is all left to the user. In this paper we are only reporting our practical clinical method for processing, displaying, and reviewing—in short, evaluating—the post-BgRT-delivery dose data provided by the RefleXion system. Our workflow is an attempt to make the best possible use of this dose information, which we know to be an imperfect description of reality but which can nonetheless be helpful in identifyng changes in the patient. As such, we have removed all instances of “analysis” in the manuscript and replaced them with more appropriate specifiers (“evaluation,” “method,” etc.). We believe this aspect of our revisions will address many of the questions that arose on initial review.
We have also made extensive revisions to the text in an effort to make it clearer and more understandable to a general readership.
Below we respond to comments point by point.
[2nd review response] First, the explanation of the scaled-dose evaluation could be expanded slightly. Although the concept is already described, adding a brief clarification on why the scaled dose is particularly useful for detecting fraction-to-fraction variations would help readers who may be less familiar with this approach. For example, in Section 2.4 you might consider modifying the sentence “A cumulative dose is not as sensitive to delivery changes as a scaled dose distribution because the latter magnifies variations that occurred in a single treatment session” to something along the lines of “Scaled dose distributions were especially useful for monitoring delivery consistency, as they amplify even small session-to-session variations that may otherwise be obscured when examining cumulative dose.”
[2nd author response] We thank the reviewer for this recommendation. We have modified the text to incorporate a variation on the suggested phrasing.
[2nd review response] Second, the discussion regarding the difficulty of interpreting target dose could benefit from a short connecting remark about potential surrogate indicators, such as the biology tracking zone (BTZ). You already elaborate on this concept later in the manuscript; introducing it briefly earlier in Section 4.2 would improve continuity. For instance, after noting the limitations of interpreting dose to a moving target, you could add: “Although direct interpretation of target DVHs is challenging under these conditions, surrogate indicators such as the dose distribution within the biology tracking zone (BTZ) may offer complementary insight and merit further investigation.”
[2nd author response] We are reluctant to bring up the BTZ at an early juncture, as it is a technical concept requiring some explanation, which would be a distraction in this portion of the text. We added the sentence “It is conceivable a static structure could be defined that would serve as a surrogate indicator for the dose received by a moving BgRT target, but this is a topic requiring investigation.”
[2nd review response] The readability of the trend plots in Figures 3 and 4 could also be enhanced. The information they present is valuable, but the current contrast and density make them somewhat difficult to interpret quickly. Clarifying in the legend that the red region corresponds to tolerance violations, or enlarging the labels for the plotted metrics, would improve usability.
[2nd author response] We have modified the trend plots as requested.
[2nd review response] A few sentences could be tightened for clarity. In the Abstract, the sentence “Our workflow makes efficient use of existing RefleXion and in-house software resources” might be clearer if revised to: “The workflow integrates existing RefleXion tools with our in-house software to provide an efficient framework for post-delivery dose evaluation.”
[2nd author response] We fear the suggested phrasing is not quite what we intend to communicate here, for several reasons. We would definitely not claim that our workflow is an “efficient framework,” as we even explicitly acknowledge in the Discussion/Future directions section that it is quite labor intensive. We are also reluctant to describe the combined qualitative+quantitative dose review as a “framework” because we have exclusively used that term in the manuscript to refer to our custom dose-volume metric calculation apparatus. And we are reluctant to characterize our workflow as “integrating” RefleXion and in-house software tools because that suggests to us an integration of systems, which is not the case.
What we mean to convey with “our workflow makes efficient use of existing […] resources” is that we are making maximal use of existing resources, rather than re-creating things from scratch. We recognize from the reviewer’s suggestion, though, that the gist of our original phrasing was not entirely clear. We have replaced it with “Our workflow efficiently incorporates existing RefleXion TPS features and in-house software into a process for thoroughly evaluating doses to OARs after BgRT delivery.”
[2nd review response] Similarly, in Section 3, the informal expression “nothing attention-grabbing” could be replaced with the more formal “no prominent deviations were observed.”
[2nd author response] We agree the previous phrasing was too colloquial and have replaced it with “the deviations were not prominent.”
[2nd review response] Lastly, I suggest considering the following revised title, which more clearly reflects the focus and contribution of the work: “A Clinical Workflow for Post-Delivery OAR Dose Evaluation in Biology-Guided Radiotherapy (BgRT)”
[2nd author response] We have modified the title to explicitly indicate the focus of this paper is on “a clinical workflow.”
Reviewer 4 Report
Comments and Suggestions for Authors
Lines 126, 178, 187, 199, 200, 207, etc. throughout.: I notice throughout that the authors cite each case as what looks like the institutional nomenclature for ROIs in their TPS. I can readily guess what each one is, that LungLUL5500_5 is a left upper lobe lung tumor treated with 5500 cGy in 5 fractions; however, for a general readership, it is best to describe it in full. Of course, common acronyms like LUL are fine, it is just stylistically better to avoid using the institutional ROI coding (_underscores and all) and write out the description in full.
We thank the reviewer for pointing this out. We are so habituated to our institutional nomenclature that we failed to recognize it would be unfamiliar to the casual reader. We have removed this terminology and described the dose fractionation in full.
2nd review response: Thank you.
Lines 135-137: Please add some details to your three clinical cases somewhere in the “Materials and Methods” section. I realize that there’s some detail scattered throughout the paper, but there should be 1-2 paragraphs on Case description in the “Methods” section. Suggestions on what this content might look like:
Where are the targets (bone can be anywhere, it matters where in the lung the targets are—mediastinal? lower lobes?);
what are the prescriptions and fractionation?;
what exactly do you mean by “uncomplicated”?; what do you mean by “presented difficulties during treatment”?
do such difficulties bound what is “complicated” and what is “uncomplicated”?
what is the primary disease?
I think that detail would give the reader better perspective as to what is being treated. E.g. there might be a case that is a lower left lung tumor, 3 cm in diameter, treated with an SBRT dose of 52 Gy in 4 fractions, of some pathology, and was complex as the target had a large motion trajectory of 3 cm due to proximity to the diaphragm. You can see how that level of detail can give the reader an appreciation of the clinical scenario, and even sell the reader on why the BgRT motion-compensation platform coupled with your novel workflow is clinically useful. Related to my previous comment on nomenclature: if you stated in the “Methods” what Case 1, Case 2, and Case 3 is, then you can refer throughout the paper to each Case#, rather than the coded institutional ROI nomenclature that is used throughout.
We thank the reviewer for these helpful recommendations. We agree that labeling the retrospectively analyzed patient cases by number simplifies their presentation, and we have implemented that recommendation. We have also added more information about each case, per the suggestions above. We eliminated the terms “complicated/uncomplicated” and now simply state whether each treatment was delivered to completion or not.
2nd review response: Thank you. In combination with eliminating the institutional ROI nomenclature, this very much improves readability.
Lines 130-137, 2.4 Custom dose analysis workflow: In-house software based on the Varian environment is introduced here, meant to “process the dose information”. Considering that this is the key value-added feature that the authors are presenting (the title is “Evaluation of dose to organs at risk after BgRT…” after all), I believe more elaboration is required at this point. I realize that this elaboration of the in-house workflow/software occurs in the next paragraphs, which results in the next comment…
Lines 139-176: These paragraphs describe the qualitative evaluation using the built-in tools in the RefleXion TPS, and the quantitative evaluation using the Eclipse-based software. These paragraphs I believe belong not in the “Results” section, but rather in the “Materials and Methods” sections, specifically 2.3 and 2.4 (recall the immediately previous comment where it looked like detail on the in-house software was missing). While it is reasonable for some amount of “Methods” language to be included in the “Results” section (e.g. we weren’t expecting this quirk in this experiment while taking measurements, so we’ll describe it while showing the data), Lines 139-176 are very much describing the general method, and shouldn’t be in “Results”. This is a simple fix: just move these paragraphs into “Methods”, with some wordsmithing to smooth it over.
We agree this is a more natural organization of the material, and we have implemented the reviewer’s suggestion. We moved a significant chunk of text from “Results” to “Methods,” and we modified/added subsections in “Methods” to better organize the information presented there. We believe these extensive changes have improved the explanation of the in-house components of our workflow, as requested in the comment re: lines 130-137.
2nd review response: This improves the readability, thank you.
Figure 2: The text in the figure is far too small. A general rule of thumb is that the smallest text in a figure should be within at least ~3 points of font size of the main text; however, in Figure 2, the smallest text size is by my measurement 9 points of font size smaller than the main text. Even though this is a digital publication, the reader should be able to read at normal zoom levels. Moreover, Figure 2 is pixelated when zoomed in to the 500% required to read the smallest text, which further underlines that the text really should be reformatted. I realize that this is a screenshot of a TPS-related application, but there are ways of displaying this info without having such a small font size.
We thank the reviewer for bringing this resolution issue to our attention. We have recreated the figure (which is now Figure 3) with better resolution. If this is unsatisfactory we will work with the editors to find a solution to improve its readability.
2nd review response: I leave it to the editor as to whether this is an acceptable font size and resolution.
Figure 2 and 3: Both figures show LUL lung cases, yet Figure 2 shows 11 OAR metric trend lines while Figure 3 shows the “problem” lung case with only 3 OAR metric trend lines. As both are in the upper lung, you would expect the list of OAR metrics in play to be broadly similar, but they are not—why is this? If the complete OAR metric list was shown in Figure 3, would it obscure some of that upward trend, or add to it?
We thank the reviewer for pointing out this inconsistency in our presentation of the two LungLUL cases. For the problem case (patient case 3) we originally only graphed the three dose-volume metrics which cross into the red violation region, because that is the behavior that would have triggered an investigation. However, the reviewer is absolutely correct that for the sake of consistency we should include the same metrics that are graphed for LungLUL patient case 2. We have recreated the trend plots for patient cases 2 and 3, and we have applied a consistent labeling scheme for the metrics in both.
2nd review response: Thank you. This enables a proper comparison between Cases 2 and 3.
Lines 175-176: “that are ostensibly exceeding their dose-volume tolerances or trending in that direction.” What does “ostensibly” mean? Either tolerances are breached or they are not. What does “or trending in that direction” mean? How many fractions make a trend? How many OAR metrics make a trend that is a potential problem? How large does the variation have to be to make a trend that raises a flag?
Re: “ostensibly”: We used this qualifier to reflect the fact we have no way of knowing the accuracy of the calculated OAR dose-volume metrics, because BgRT is a dynamic process that responds to target motion but the post-BgRT dose computation is performed on a static CT image. It is possible, therefore, that a calculated OAR dose-volume metric may show a tolerance violation even if in reality the OAR did not actually receive such a dose—ie, it could be a false positive. (The converse could happen as well, a false negative.) However, we acknowledge that using the term “ostensibly” may only create confusion, so we have reworded the relevant passages in the abstract and text to simply state, eg, “OARs whose calculated metrics exceed tolerance limits.”
The judgment of what constitutes a concerning OAR dose violation or trend is left entirely to the user. We have added text to explicitly state that we are not providing prescriptive guidance on interpreting the numerical results from our workflow. (For me, any new tolerance violation, or a two-fraction trend approaching/crossing the tolerance limit, would be an investigation trigger, but that is solely my own opinion.)
2nd review response: Please see my overall comments at the end. In your comment above, it is some amalgam of my suggested remedies (a) and (c).
Line 221: It is difficult to follow which lung case is being referred to here, as there are two lung cases. It would be far easier to follow along throughout if each of Case 1, Case 2, and Case 3 were described in a section in “Materials and Methods” (as I recommended previously), and then throughout the rest of the paper you can refer to each case by number.
We agree this would simplify the presentation of the retrospectively evaluated cases, and we have implemented the reviewer’s recommendation.
2nd review response: Thank you.
Lines 221-233: The claim is “our method would have readily flagged” an issue with one of the lung cases. This paragraph then describes how the case was flagged: viewing the kVCT which indicated shrinkage, and follow-up review of the bDVHs. Both of these methods use features that are already in the RefleXion TPS application. Thus I do not see how the authors’ claim is valid, i.e. that their method would have caught this issue, if the features already in the TPS were adequate for screening. I take issue with the statements “for most patients it is more likely that changes in post-BgRT OAR doses could warn of unnoticed changes in the target’s anatomy or PET activity which might lead to toxicities or affect future treatment sessions”, this is a broad claim not supported by data in the article. The claim “our method would have raised warning flags as early as fraction 1, before there was a visible change in tumor size” is not factually supported in the text—how would it have raised a flag, and what part of your method would have flagged it? I might assume that it refers to the upward trend in Figure 3? If so, I do not see much difference in the % increase of the rib OAR metrics between Plan and Fx1 for the two lung cases in Fig 2 and 3—if this is the warning flag that the authors are citing, what’s the difference between the case where it delivered and the case where the course didn’t complete? The rib doses went up between Plan and Fx 1 for both lung cases, and had at least one rib metric in the red zone for Fx 1—wouldn’t this flag a problem for both lung cases?
We thank the reviewer for pointing out some sloppiness in our arguments here. The reviewer is correct that we were making some broad claims (eg, “for most patients[…],” “our method would have raised warning flags as early as fraction 1”) that are not supported by the evidence presented in the manuscript. We have revised our wording in those cases to be more accurate.
Re: the lung patient with shrinking tumor (case 3): the reviewer is correct that the shrinkage was noticed and evaluated using existing features of the RefleXion TPS. However, this was thanks to the astuteness and initiative of the supervising physicist, who noticed the physical change and decided to perform a manual investigation and calculations which revealed a violation of the brachial plexus Dmax had occurred. With our new workflow, these calculations can be performed and displayed for the entire suite of custom OAR metrics of interest, and there is far less risk of missing a violation or trend because of observer variation. We recognize that the RefleXion display does have value, though, which is why our workflow consists of both a qualitative review in the RefleXion TPS and a quantitative review with our in-house software.
Regarding the OAR trend plots for the two lung cases: the reviewer is correct that the two-fraction trend of increasing doses in case 3, plus the entry of three separate metrics into the red violation zone, would have been a clear indication to us that further investigation was warranted. In contrast, with case 2 no metrics entered the red violation zone that were not already there in the original treatment plan. However, the reviewer is also correct that there is potentially some element of coincidence to the fact that the case which encountered delivery issues is also the one in which OAR metrics transitioned into the red violation region. One can imagine a scenario in which a metric started in the violation region and kept trending worse and worse. The situation will vary case to case, which is why we leave interpretation to the user, and why it is hard to present a comprehensive algorithmic solution. Our goal here is to take potentially useful dose information which might otherwise go unused and present it in an efficient manner.
2nd review response: I believe that this is the main problem that I have, that the trigger mechanism is left entirely to the user’s interpretation. If it were my interpretation, I would consider an OAR already in the red zone, trending higher by 20%-30% to be alarming (as in Case 2, Figure 3b, from Plan->Fx1 and Plan->Fx4), and yet the article and the authors’ comment above appear to dismiss that circumstance. Please see my Overall Comments at the end for suggested remedies.
Lines 239-243: This is the first mention of machine log files in the paper, but I suppose it was implied earlier on when calculating according to the post-BgRT “treatment record” (Line 107). I think mention of the term “machine log files” would be helpful at Line 107, as it would link up with the later mention in the Discussion.
We thank the reviewer for this suggestion, which has been implemented.
2nd review response: Thank you.
Lines 244-253: “We are less concerned with the absolute accuracy of OAR delivered-dose values… and are instead more focused on variations which could indicate that functional or anatomical changes have occurred in the target region and warrant closer scrutiny”; and Lines 182-183: “make it easy to recognize if a particular OAR is violating its nominal tolerance limits and/or if its situation is worsening.” As with any QA/screening tool, there must be thresholds in order to enable the decision-making—the above quotes from the paper are quite ambiguous, it seems you are saying “both absolute dose accuracy and relative variation, one more than the other”. In the context of your work, if variations truly are the important thing (as you say in 244-253), you might set, say, a +20% increase relative to the reference plan, above which you would further investigate and below which you wouldn’t. Also, how many OARs need to raise a flag before you investigate? The claim is this is a “quantitative, graphical analysis of OAR dose metrics” (Lines 317-318)—such an analysis is incomplete without thresholds, and an algorithm to interpret the thresholds if there’s more than one. To recap, this is ambiguous: are the thresholds based on absolute OAR doses i.e. the “red zones”, or based on variation like is claimed in Lines 244-253, and if both, in what proportion, and navigated by what algorithm?
Lines 235-237: “If any concerning OAR dose deviations or trends…” Again, echoing my previous comment, this sounds like decision-making is based on “magnitude of concern”, which isn’t quantitative. Specific, threshold-based, decision-making criteria are needed to flag issues. Again, I understand that you have the “red zones” displayed in Figures 2 and 3; however, you also state that you are “more focused” (Line 249) on % increase of OAR metrics—these all need thresholds and a governing algorithm, since you are claiming this is a quantitative method.
We thank the reviewer for pointing out that our terminology here was confusing. As discussed in some of our responses above, this manuscript is only reporting the procedure we developed for calculating and displaying custom dose-volume metrics after BgRT delivery, and how review of that information could be clinically useful when applied to OARs. We therefore agree that our use of the word “flag” was misleading, as it suggests application of an algorithm with decision thresholds, which is not the case. We have removed all instances of “flag”/”flagging” and reworded the surrounding text.
We feel it is still accurate, however, to refer to our workflow as “quantitative,” because we are calculating dose-volume metrics from the RefleXion-provided post-delivery dose distribution. We also feel it is appropriate to provide some modest commentary on what we ourselves consider important when we review the dose data clinically, in light of its limitations/shortcomings. But ultimately all judgments are up to the user.
We have revamped much of the Discussion text to clarify these many points. We thank the reviewer again for their feedback, which has resulted in vast improvements to the manuscript.
2nd review response: Please see my Overall Comments at the end.
Discussion: Only one of the lung cases was discussed, what of the other two cases? Shouldn’t there be interpretation applied to all the data in the Discussion? The Discussion section should link back to the data/observations in the “Results” section.
We considered the workflow results for the other two cases largely unremarkable, which is why we stated that our retrospective investigation “supports our clinical experience that for most BgRT patients the dose is delivered as intended.” We felt it would be more illuminating to discuss the motivation behind our workflow (i.e., case 3) as well as the goals and potential benefits of a custom dose-evaluation framework.
2nd review response: I respectfully disagree that the other two cases should not be discussed in Discussion. If the claim was that this is a quantitative tool to catch BgRT issues as the treatment course progresses, you would need to comment how the data from Cases 1 and 2 contrasts with the data from Case 3. At the very least, Case 2 does exhibit what might be even more alarming excursions into the red zone, which I expand on in my New comment on Figure 3, and in Overall comments below.
-----------------------------------
New comments:
I believe Figure 2 is new, and does help in describing the workflow. It would be helpful for the reader to understand why some of the text is bolded blue in the figure. The best place for this description would be in the Figure caption.
Line 191: What is a “significant offset”? Would that be exceeding a PTV margin?
Line 205: “significantly worsened”. This is non-quantitative. How much worsening before it is a concern?
Figure 3: In conjunction with the previous comment, as well as my previous comments: in Case 2, there are 3 OARs already in the red zone that all increase from Plan->Fx1, with a rib metric increasing by 0.2, which seems a lot. Also, comparing Case 2, Plan->Fx4, there is an even more alarming rise in that same rib metric. Yet, this is not remarked upon in the Results or Discussion. In fact, it could be interpreted that line 237 “ultimately there were no worsening trends of clinical concern” contradicts what is visible in the Plan->Fx1 and Plan->Fx4 increases. Wouldn’t OARs already in the red zone going further into the red zone be more alarming than an OAR crossing from “safe” to red zone? Even if the clinician agreed at the Plan stage that, say, rib dose 40% over is acceptable, perhaps 60%-70% is not? I understand the modifier “of clinical concern” is there, but isn’t some additional remark needed to say how these rather large increases were evaluated and eventually the decision was made to continue with treatment? Finally, isn’t part of this OAR metric analysis to catch if there’s a PTV size or shape difference, as in Case 3? I.e. that part of this tool isn’t just to catch OAR doses straying out of tolerance, but to catch large anatomical changes in the target, as is claimed in the Discussion on Case 3 (lines 278-289)? If that is the case, would it not be a trigger for target-size-change if an OAR already in the red zone strays further into the red zone?
Overall comments:
The authors present a visualization tool to evaluate OAR doses for a delivered fraction, which appears to fill a need that the RefleXion software does not provide; however, the authors make a claim that the workflow can detect treatment problems such as target shrinkage (as in Case 3) that I do not think is supported by the data. The data from Case 2 and 3 are in contradiction with respect to this claim. I suggest 3 remedies to resolve the apparent contradiction:
- (a) Frame this article as this visualization tool *only*, without the claim that it can detect tumor shrinkage as in Case 3, as the Case 2 data directly contradicts this claim.
- (b) Alternatively, can it be explained better why Case 2 did not have a treatment intervention/inspection yet had more alarming OAR data? Suggesting that it is acceptable to have a rise in OAR trends as long as it starts in the red zone is counter to clinical practice. Whether or not you can make this explanation, then please consider (c):
- (c) The work can be framed as a screening tool, such that it over-sensitively picks up OAR differences, and in some cases (like Case 2) all is well upon inspection. In other words, perhaps it is acceptable to have false positives as long as you have sensitive (perhaps overly-sensitive) detection. But to claim this, you would either (i) require more data OR (ii) concede that at this stage, the necessary data does not exist and will have to be a future study, including thresholding and perhaps an analysis tool such as a receiver-operator curve as is typical of other screening tools.
Author Response
[2nd author response] First we want to present some introductory remarks that we meant to include in our first response but which were inadvertantly omitted. These comments are still germane to our latest responses:
We wish to thank the reviewers for their careful reading of our manuscript and for their insightful comments. Their feedback made us realize we needed to firm up our report in many places and use clearer and more precise language throughout. We consider the revised manuscript to be a vast improvement on the original submission.
A number of comments from multiple reviewers indicated to us that our terminology may have created some underlying confusion about the nature and goals of our work. In particular, our use of the term “analysis” to describe our workflow may have been misleading. To be clear, we are not providing any prescriptive guidance on how to interpret post-delivery dose-volume metrics, nor are we providing any action thresholds—that is all left to the user. In this paper we are only reporting our practical clinical method for processing, displaying, and reviewing—in short, evaluating—the post-BgRT-delivery dose data provided by the RefleXion system. Our workflow is an attempt to make the best possible use of this dose information, which we know to be an imperfect description of reality but which can nonetheless be helpful in identifyng changes in the patient. As such, we have removed all instances of “analysis” in the manuscript and replaced them with more appropriate specifiers (“evaluation,” “method,” etc.). We believe this aspect of our revisions will address many of the questions that arose on initial review.
We have also made extensive revisions to the text in an effort to make it clearer and more understandable to a general readership.
Below we respond to comments point by point.
[2nd review response] I believe Figure 2 is new, and does help in describing the workflow. It would be helpful for the reader to understand why some of the text is bolded blue in the figure. The best place for this description would be in the Figure caption.
[2nd author response] Yes, this diagram was requested by multiple reviewers in round 1 and we agree it is helpful to provide a graphical representation of the workflow. We added text to the caption to communicate that “The data review steps are boldfaced.”
[2nd review response] Line 191: What is a “significant offset”? Would that be exceeding a PTV margin? Line 205: “significantly worsened”. This is non-quantitative. How much worsening before it is a concern?
[2nd author response] These are fair criticisms. Our intent here is only to communicate the process and its purpose; as we have mentioned elsewhere, we are not providing an algorithm and all interpretation and judgments are left to the user. We have therefore changed to language to “noticeable offset” and “worsened,” respectively.
[2nd review response] I believe that this is the main problem that I have, that the trigger mechanism is left entirely to the user’s interpretation. If it were my interpretation, I would consider an OAR already in the red zone, trending higher by 20%-30% to be alarming (as in Case 2, Figure 3b, from Plan->Fx1 and Plan->Fx4), and yet the article and the authors’ comment above appear to dismiss that circumstance. Please see my Overall Comments at the end for suggested remedies.
[2nd author response] We respectfully disagree with the contention that we have been dismissive of the variations in Case 2 OAR metrics, as we explicitly added text to the Results section in our first revision to address this legitimate critique from the reviewers. (“For case 2 there were several OAR metrics that exhibited tolerance violations in the original treatment plan accepted by the physician[…] The subsequent variations in these OAR metrics over the course of treatment could have been due to modest changes that were visible in the size and shape of the lung tumor, but we did not identify any clear correlation and ultimately there were no worsening trends of clinical concern.”) Per our procedure, we (1) searched for changes that might explain the variations but did not find any, and (2) determined the violations were not of clinical concern for the physician. In this case the high-value violation the reviewer is referring to was in the Ribs V45Gy<5cc metric, which varied between 6.7cc and 8.5cc over the course of Tx. These metrics have conservative tolerances and are subject to the clinical judgment of the physician, and for this patient he did not regard the rib as a hard constraint. We feel the added text succinctly encapsulates the situation without getting sidetracked into technical details.
[2nd review response] I respectfully disagree that the other two cases should not be discussed in Discussion. If the claim was that this is a quantitative tool to catch BgRT issues as the treatment course progresses, you would need to comment how the data from Cases 1 and 2 contrasts with the data from Case 3. At the very least, Case 2 does exhibit what might be even more alarming excursions into the red zone, which I expand on in my New comment on Figure 3, and in Overall comments below.
[2nd author response] This is a fair critique. We have added a short section to the Discussion addressing Cases 1 & 2. On a related note, for Case 3 we added text to the Results explaining that brachial plexus toxicity would have been clinical concerning to the physician.
[2nd review response] Figure 3: In conjunction with the previous comment, as well as my previous comments: in Case 2, there are 3 OARs already in the red zone that all increase from Plan->Fx1, with a rib metric increasing by 0.2, which seems a lot. Also, comparing Case 2, Plan->Fx4, there is an even more alarming rise in that same rib metric. Yet, this is not remarked upon in the Results or Discussion. In fact, it could be interpreted that line 237 “ultimately there were no worsening trends of clinical concern” contradicts what is visible in the Plan->Fx1 and Plan->Fx4 increases. Wouldn’t OARs already in the red zone going further into the red zone be more alarming than an OAR crossing from “safe” to red zone? Even if the clinician agreed at the Plan stage that, say, rib dose 40% over is acceptable, perhaps 60%-70% is not? I understand the modifier “of clinical concern” is there, but isn’t some additional remark needed to say how these rather large increases were evaluated and eventually the decision was made to continue with treatment? Finally, isn’t part of this OAR metric analysis to catch if there’s a PTV size or shape difference, as in Case 3? I.e. that part of this tool isn’t just to catch OAR doses straying out of tolerance, but to catch large anatomical changes in the target, as is claimed in the Discussion on Case 3 (lines 278-289)? If that is the case, would it not be a trigger for target-size-change if an OAR already in the red zone strays further into the red zone?
[2nd author response] We want to emphasize that the patient analyses presented in this paper were retrospective in nature and therefore there was no possibility for a decision to be “made to continue with treatment.” We feel most of the questions here are addressed in our two preceding responses.
[2nd review response] The authors present a visualization tool to evaluate OAR doses for a delivered fraction, which appears to fill a need that the RefleXion software does not provide; however, the authors make a claim that the workflow can detect treatment problems such as target shrinkage (as in Case 3) that I do not think is supported by the data. The data from Case 2 and 3 are in contradiction with respect to this claim. I suggest 3 remedies to resolve the apparent contradiction:
(a) Frame this article as this visualization tool *only*, without the claim that it can detect tumor shrinkage as in Case 3, as the Case 2 data directly contradicts this claim.
(b) Alternatively, can it be explained better why Case 2 did not have a treatment intervention/inspection yet had more alarming OAR data? Suggesting that it is acceptable to have a rise in OAR trends as long as it starts in the red zone is counter to clinical practice. Whether or not you can make this explanation, then please consider (c):
(c) The work can be framed as a screening tool, such that it over-sensitively picks up OAR differences, and in some cases (like Case 2) all is well upon inspection. In other words, perhaps it is acceptable to have false positives as long as you have sensitive (perhaps overly-sensitive) detection. But to claim this, you would either (i) require more data OR (ii) concede that at this stage, the necessary data does not exist and will have to be a future study, including thresholding and perhaps an analysis tool such as a receiver-operator curve as is typical of other screening tools.
[2nd author response] We feel that (c) exactly describes the work we are reporting here, but we do not believe any caveats are necessary beyond those already presented in the paper. We have been careful throughout to not claim our approach is determinative or 100% reliable in identifying changes relevant to BgRT--all of our statements are properly qualified (“can be useful”, “potentially revealing”, “could be helpful”). Whether a patient’s OAR metrics can reveal a change of clinical significance will depend on a number of factors such as the relative location of the target and OAR, the size of the OAR, the treatment prescription, the dose-volume metric under consideration, etc. We also want to emphasize that our tool is not solely for detecting “tumor shrinkage” but (as we state in the paper) could potentially detect intrafraction patient shifts or changes in the target’s functional properties, if those affect OAR dose.
This is an admittedly imperfect screening tool, but we feel an imperfect tool—applied to a small sample of patients by way of demonstration—is better than no tool. Generating an ROC curve is not practicable given the enormous parameter space involved and the sparse number of BgRT patients. And this is a clinical work in progress, with many avenues for improvement, as we enumerate in the Future directions section.